

# Asymmetric Bethe Ansatz

Steven G. Jackson[1], Hélène Perrin[2], Gregory E. Astrakharchik[3] and Maxim Olshanii[4*]

**1** Department of Mathematics, University of Massachusetts Boston,
Boston Massachusetts 02125, USA
**2** Laboratoire de physique des lasers, CNRS, Université Paris 13, Sorbonne Paris Cité,
99 avenue J.-B. Clément, F-93430 Villetaneuse, France
**3** Departament de Física, Universitat Politècnica de Catalunya, E08034 Barcelona, Spain
**4** Department of Physics, University of Massachusetts Boston,
Boston Massachusetts 02125, USA

⋆ maxim.olchanyi@umb.edu

## Abstract

The recently proposed exact quantum solution for two $\delta$-function-interacting particles with a mass-ratio $3:1$ in a hard-wall box [Y. Liu, F. Qi, Y. Zhang and S. Chen, iScience 22, 181 (2019)] violates the conventional necessary condition for a Bethe Ansatz integrability, the condition being that the system must be reducible to a superposition of semi-transparent mirrors that is invariant under all the reflections it generates. In this article, we found a way to relax this condition: some of the semi-transparent mirrors of a known self-invariant mirror superposition can be replaced by the perfectly reflecting ones, thus breaking the self-invariance. The proposed name for the method is *asymmetric Bethe Ansatz* (asymmetric BA). As a worked example, we study in detail the bound states of the nominally non-integrable system comprised of a bosonic dimer in a $\delta$-well. Finally, we show that the exact solution of the Liu-Qi-Zhang-Chen problem is a particular instance of the the asymmetric BA.

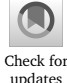

## Contents

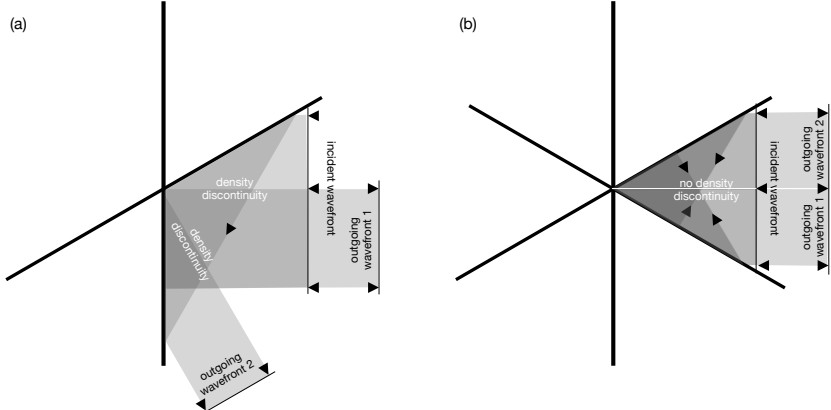

Figure 1: **A comparison between (a) a generic and (b) a Bethe Ansatz integrable systems of mirrors.**

# 1 Introduction

## 1.1 Bethe Ansatz and glimpses of its asymmetric extension

Despite its name, Bethe Ansatz [1] is a more rare phenomenon than a broadly applicable method. Unlike a variational Ansatz, which allows for setting a rigorous upper bound for the system's energy but never gives its exact value (see for example a mean-field Ansatz for the wavefunction of any interacting Bose gas, in any trap, and in any number of spatial dimensions [2]), the Bethe Ansatz attempts at producing exact answers (in a limited setting: typically short-range-interacting gases in a flat background, in one dimension). In brief, the Bethe Ansatz asserts that the eigenstates of a given many-body system is a superposition of a *finite* number of multidimensional plane waves with piece-wise-constant coefficients. It is however remarkable how broad the applicability of the Bethe Ansatz is [3], given this extraordinarily constraining demand.

To illustrate how extraordinary rare the situation is when a multidimensional wave function consists of a finite number of plane waves, consider the example of a two-dimensional plane wave scattering off a 120° wedge of reflective mirrors (Fig. 1(a)). One can follow the wave as it is being sequentially reflected from the mirrors. Let us first look at the probability density, neglecting the interference effects. The presence of a junction between the two mirrors leads to two density discontinuities. Reassigning phases to the waves will not cure discontinuities in the time and space averaged density. Under a full quantization, the density discontinuities will lead to *continuously many* new momenta. At this moment, it becomes clear that no Bethe Ansatz is possible in this setting.

Let us now add the third mirror, at 60° to the previous two (Fig. 1(b)). Curiously, the density discontinuities disappear. One can show that even when the phases are resigned, the wavefunction remains continuous. In fact, a naïve superposition of the plane waves that we have just built constitutes an *exact* solution for the problem. This is an instance of the Bethe Ansatz.

Chapter 5 of the book [1, Sec. 5.2] addresses the question of what a sufficient condition is for a system of mirrors—including the semi-transparent mirrors generated by $\delta$-functional hyperplanes — to be solvable using the Bethe Ansatz. The answer is as follows: the system of mirrors in question (including the assignment of the $\delta$-functions' strengths) must form a *multidimensional generalized kaleidoscope*, i.e. it must be *invariant under any sequence of reflections generated by the mirrors themselves*. In our article, we are attempt to advance beyond this paradigm. We will call this method an asymmetric Bethe Ansatz (asymmetric BA).

## 1.2 Multidimensional kaleidoscopes

The concept of a *generalized kaleidoscope* is the key to understanding the mathematical nature of Bethe Ansatz solvability for some systems of $\delta$-function-interacting particles [1, Sec. 5.2]. It is a generalization of a conventional notion of a kaleidoscope to kaleidoscopic systems of, generally, semitransparent mirrors.

A conventional kaleidoscope is a system non-transparent mirrors that is invariant under all reflections generated by its own mirrors. As a consequence of this invariance, images of objects in a kaleidoscope are not distorted at the junctions between the mirrors, rendering these junctions invisible for an observer.

Transformations of space generated by all possible sequences of reflections about mirrors of a kaleidoscope form a *reflection group* [4].

All indecomposable (to products of reflection subgroups) reflection groups are known. They are catalogued using the so-called Coxeter diagrams [5]; the latter are shown in Figs. 2-3. Decomposable reflection groups can be factorized onto products of the indecomposable ones.

Figure 2 lists the so-called indecomposable *finite* reflection groups. For a finite reflection group, all of its mirrors cross the origin. The mirrors divide the space onto the infinite volume *chambers*. Each of the chambers is connected to any other via an element of the corresponding reflection group. Reflections about chamber's mirrors can be used to generate the whole reflection group.

For an indecomposable finite reflection group, the number of mirrors that form its chamber equals the number of spatial dimensions of the space in which the group operates: as a result, the dihedral angles between its mirrors fully determine the corresponding kaleidoscope. In Fig. 2, circles represent the mirrors of a chamber. The angle between two mirrors whose circles are not directly connected by an edge is $\pi/2$. Edges with no label correspond to an angle $\pi/3$ between the mirrors. Otherwise, an index $m$ on an edge would give an angle $\pi/m$.

Figure 3 catalogues the so-called indecomposable *affine* reflection groups. There, the mirrors divide the space into finite-volume *alcoves*. The number of mirrors that bound an alcove exceeds the number of the spatial dimensions by one. As the result alcoves are closed simplexes. Similarly to the case of indecomposable finite reflection groups, each of the alcoves is again connected to any other via an element of the corresponding reflection group; reflections about the walls of any given alcove generate the full reflection group.

In the affine case, mirrors are arranged in space into periodic series. Each affine group has a finite partner (the converse is not generally true): a union of images of an alcove produced by the corresponding finite group forms an elementary periodicity cell, whose periodic translations tile the space.

Figure 3 lists the indecomposable affine reflection groups. Each Coxeter diagram encodes the dihedral angles between the mirrors of the alcove of the group. Here, notations are identical to the ones for the finite groups. However, unlike in the finite case, dihedral angles between the mirrors of an alcove determine the corresponding kaleidoscope only up to an arbitrary dilation factor.

Decomposable reflection groups produce disconnected Coxeter diagrams, one connected sub-graph for each indecomposable factor (with factors being represented by either an indecomposable finite or an indecomposable affine reflection group). For the decomposable groups, the notion of a chamber (in case of an infinite volume) or alcove (for finite volume kaleidoscopes) can be preserved without modification. Note however that when one or more of the indecomposable factors of the decomposable group are affine, the Coxeter diagram and the dihedral angles between the chamber/alcove mirrors it codifies fix the corresponding kaleidoscope only up to a set of arbitrary dilation factors, one for each indecomposable affine factor of the decomposable reflection group in question.

## 1.3 Gaudin's generalized kaleidoscopes

In his classic book [1], Michel Gaudin proposed [1, Sec. 5.2] a model that he called a *generalized kaleidoscope*. Consider a standard kaleidoscope. Let us replace some of its mirrors by $\delta$-function potentials $g_{\mathbf{n},\mathbf{r}_0}\delta(\mathbf{n}\cdot(\mathbf{r})-\mathbf{r}_0)$, with the coupling constants $g_{\mathbf{n},\mathbf{r}_0}$ chosen in such a way that the resulting system of semitransparent mirrors still respects the reflection group of the original kaleidoscope. Here, $\mathbf{n}$ is a unit normal to the mirror, $\mathbf{r}_0$ is a point on the mirror, and $g_{\mathbf{n},\mathbf{r}_0}$ is the strength of the $\delta$-function potential. The remaining non-transparent mirrors can be naturally reinterpreted as $\delta$-function potentials of an infinite strength. A generalized kaleidoscope problem is a problem of finding the eigenstates and eigenenergies for a multi-dimensional quantum particle moving in such potential. According to Ref. [1, Sec. 5.2], any generalized kaleidoscope is solvable using Bethe Ansatz.

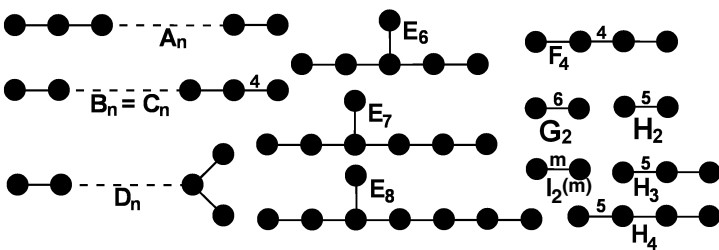

Figure 2: **A full list of indecomposable finite reflection groups, encoded by the Coxeter diagrams**. For $A_n$, $n = 0, 1\,2, \ldots$. $A_0$ coincides with the trivial group (i.e the group whose only element is the identity). For $B_n = C_n$, $n = 2, 3, 4, \ldots$. For $D_n$, $n = 4, 5, 6, \ldots$. For $I_2(m)$, $m = 5, 7, 8, 9, \ldots$

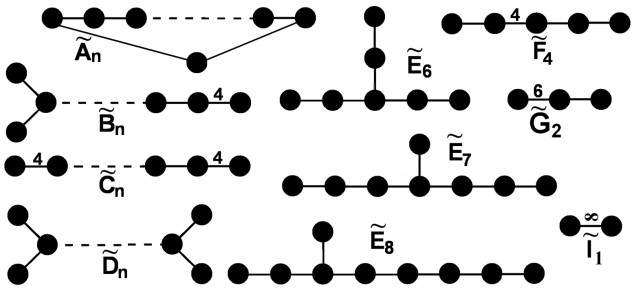

Figure 3: **A full list of indecomposable affine reflection groups, encoded by the Coxeter diagrams**. For $\tilde{A}_n$, $n = 2, 3, 4, \ldots$. Here $\tilde{I}_1$ corresponds to two parallel mirrors as an alcove. Note that in some texts, $\tilde{A}_1$ is used instead of $\tilde{I}_1$. For $\tilde{B}_n$, $n = 3, 4, 5, \ldots$. For $\tilde{C}_n$, $n = 2, 3, 4, \ldots$. For $\tilde{D}_n$, $n = 4, 5, 6, \ldots$. For $I_2(m)$, $m = 5, 7, 8, 9, \ldots$

The generalized kaleidoscope model is a natural generalization of the three textbook Bethe-Ansatz-solvable, empirically relevant models: $N$ $\delta$-potential-interacting spinless bosons [6] or spin-1/2 fermions [7, 8] on a ring (reflection group $\tilde{A}_{N-1}$), and $N$ $\delta$-potential-interacting bosons in a hard-wall box (reflection group $\tilde{C}_N$) [8].

One particular consequence of the reflection symmetry of any generalized kaleidoscope is as follows:

**Integrability Condition 1 (Standard necessary condition for Bethe Ansatz integrability)**
*If a system of $\delta$-function mirrors is Bethe-Ansatz-integrable, then any two of its $\delta$-function that cross at a dihedral angle $\pi/$(odd number) carry the same coupling constant.*

Indeed, for two mirror reflections $\hat{\mathcal{T}}_1$ and $\hat{\mathcal{T}}_2$ whose mirrors cross at a dihedral angle $\pi/m$, their composition

$$\mathcal{T}_1\mathcal{T}_2 = \mathcal{R}_{\frac{2\pi}{m}} \, ,$$

is a rotation by $\frac{2\pi}{m}$ in the plane spanned by the normals $\mathbf{n}_1$ and $\mathbf{n}_2$ to the $\hat{\mathcal{T}}_1$ and $\hat{\mathcal{T}}_2$ mirrors respectively, in the direction from $\mathbf{n}_1$ to $\mathbf{n}_2$. Under these conventions, $\mathbf{n}_1$ and $\mathbf{n}_2$ are related as

$$\mathbf{n}_1 = \mathcal{R}_{-\frac{\pi}{m}}\mathbf{n}_2 \, .$$

Applying the $\mathcal{T}_1\mathcal{T}_2$ transformation $\frac{m-1}{2}$ times, one gets

$$(\mathcal{T}_1\mathcal{T}_2)^{\frac{m-1}{2}} = \mathcal{R}_{\pi-\frac{\pi}{m}} \, .$$

Applying this transformation to the normal to the second mirror we get

$$(\mathcal{T}_1\mathcal{T}_2)^{\frac{m-1}{2}}\mathbf{n}_2 = \mathcal{R}_{\pi-\frac{\pi}{m}}\mathbf{n}_2 = -\mathbf{n}_1 \, .$$

Since both $\hat{\mathcal{T}}_1$ and $\hat{\mathcal{T}}_2$ are mirrors of the kaleidoscope of interest, the corresponding generalized kaleidoscope must be invariant under any element of the reflection group generated these mirrors; in particular, it must be invariant under the transformation $(\mathcal{T}_1\mathcal{T}_2)^{\frac{m-1}{2}}$. This invariance requires that the coupling strengths are equal for the two mirrors under consideration. Notice that the above construction requires $\frac{m-1}{2}$ be integer, and hence $m$ itself be odd. Fig. 4 illustrates our construction for the case $m = 5$, in two dimensions.

Recall that above, we assumed that in order to be Bethe-Ansatz-solvable solvable, a generalized kaleidoscope must be symmetric with respect to the reflection group it is generated by. The asymmetric Bethe Ansatz solvability considered below relaxes this requirement.

## 1.4 Open question: The source of integrability of the Liu-Qi-Zhang-Chen system

In Ref. [9], authors present an exact solution for a quantum system consisting of two $\delta$-interacting particles with a mass ratio $3 : 1$ in a one-dimensional hard-wall box. After a suitable change of variables, the problem reduces to a motion of a two-dimensional scalar-mass particle in a rectangular well with a width-to-height ratio $1 : \frac{1}{\sqrt{3}}$, divided by a *finite strength* $\delta$-function barrier along its grand diagonal. If the standard Bethe Ansatz had been the source for the

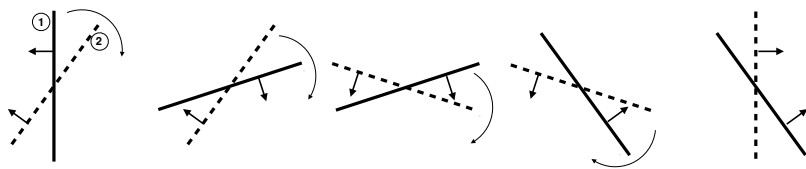

Figure 4: **A two-dimensional illustration to the Integrability Condition 1.**

solution, both the walls and the barrier would be mirrors of a kaleidoscope. However, in that case, the distribution of the coupling constants—for example a finite coupling on the diagonal and an infinite coupling on the left vertical wall, at $\pi/3$ to each other—would contradict the Integrability Condition 1.

The absence of an explanation for the existence of the exact solution to the Liu-Qi-Zhang-Chen system was our primary motivation for this project.

In this paper, our objective is to find the underlying mathematical mechanism that allows for the Liu-Qi-Zhang-Chen solution [9]. We aim to identify a general mathematical phenomenon behind it, and in doing so, to enlarge the standard Bethe Ansatz paradigm.

# 2 Asymmetric Bethe Ansatz

## 2.1 Explaining the Liu-Qi-Zhang-Chen solution and the asymmetric Bethe Ansatz

We suggest the following explanation for the existence of the exact solution of the Liu-Qi-Zhang-Chen problem [9].

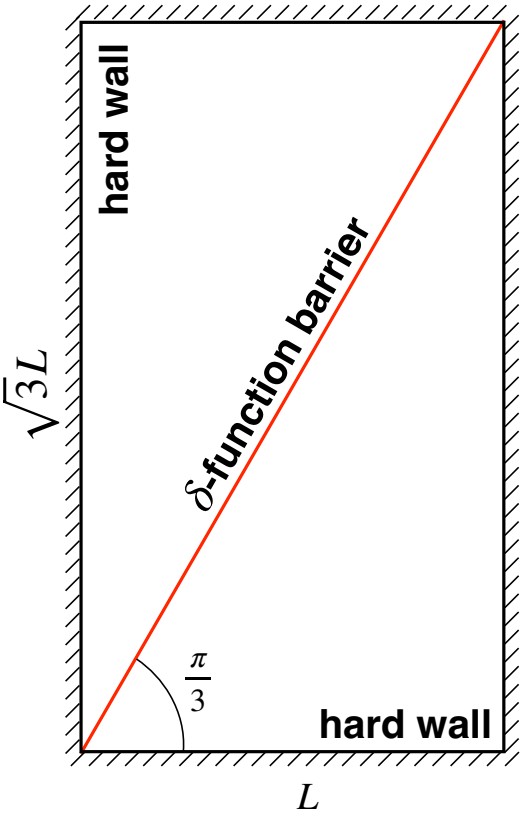

Figure 5: **A rendering of the Liu-Qi-Zhang-Chen system suitable for analysis of its integrability.** A two-dimensional scalar mass quantum particle is assumed. The original system consisted of two one-dimensional $\delta$-interacting particles ($V(x_1 - x_2) = g\delta(x_1 - x_2)$) with masses $m_1$ and $m_2 = 3m_1$, in a hard-wall box of length $L$ [9].

1. Consider a generalized kaleidoscope based on a reflection group $\mathcal{G} = \tilde{G}_2$ (see Fig. 6(a), red and blue lines). Below, we will refer to it as a $\mathcal{G}$-generated generalized kaleidoscope. Here, the red and blue lines represent mirrors of that kaleidoscope. Two mirrors, each shown with a different color, are allowed to have two different coupling constants without violating the Integrability Condition 1.

2. Now, consider a *reflection subgroup* of $\tilde{G}_2$ represented by $\mathcal{H} = \tilde{A}_1 \times \tilde{A}_1$: the mirrors of the latter are shown as the black dashed lines. Note that generally, such a subgroup is *not symmetric* with respect to the principal group $\mathcal{G}$.

3. Consider the eigenstates of the $\mathcal{G}$-generated generalized kaleidoscope that are odd with respect to each of the mirror reflections that generate $\mathcal{H}$. Such eigenstates will be, at the same time, the eigenstates of another quantum system. To build this system one starts from the $\mathcal{G}$-generated generalized kaleidoscope and replaces the mirrors coinciding with the mirrors of $\mathcal{H}$ with the infinite strength walls. Let us call it a $\mathcal{G}$-$\mathcal{H}$-generated asymmetric generalized kaleidoscope.

   Note that the new coupling constant assignment *does violate* the Integrability Condition 1. Indeed the infinite coupling "black dashed" mirrors and finite coupling "red mirrors" cross at a $\pi/3$ angle.

4. The $\mathcal{H}$-odd eigenstates of the $\mathcal{G}$-generated generalized kaleidoscope form a complete basis in the space of functions that are $\mathcal{H}$-odd. Hence the $\mathcal{H}$-odd eigenstates solve the problem of quantization of the $\mathcal{G}$-$\mathcal{H}$-generated asymmetric generalized kaleidoscope.

The Liu-Qi-Zhang-Chen system can be reinterpreted as an alcove of a $\mathcal{G}$-$\mathcal{H}$-generated asymmetric generalized kaleidoscope, with $\mathcal{G} = \tilde{G}_2$ and $\mathcal{H} = \tilde{A}_1 \times \tilde{A}_1$, for the following assignments

- The "red" mirrors of the $\mathcal{G}$-$\mathcal{H}$-generated asymmetric generalized kaleidoscope (see Fig. 6) are assigned the coupling constant along the "red" $\delta$-plate of the Liu-Qi-Zhang-Chen problem (Fig. 5).

- The "blue" mirrors of the $\mathcal{G}$-$\mathcal{H}$-generated asymmetric generalized kaleidoscope (see Fig. 6) are assumed to be transparent (i.e. assigned a zero coupling constant).[1]

- One of the "alcoves" of the reflection group $\mathcal{H}$ (a part of space bounded by its mirrors, grey rectangle at Fig. 6)(a) is interpreted as the rectangular box Liu-Qi-Zhang-Chen 2D particle is confined to.

This concludes our interpretation of the exact eigenstates of the Liu-Qi-Zhang-Chen system (Fig. 5) as an instance of applicability of an extended, "asymmetric" version of the conventional Bethe Ansatz [1, Sec. 5.2]. We suggest *asymmetric Bethe Ansatz* (asymmetric BA) as the name for this method. In the same vein, we will call the corresponding system of mirrors a *asymmetric generalized kaleidoscope*. Figure 6(b) shows the ground state of the Liu-Qi-Zhang-Chen system.

Asymmetric BA features a relaxed version of the Integrability Condition 1:

**Integrability Condition 2 (Necessary condition for the asymmetric Bethe Ansatz integrability)** *If a system of $\delta$-function mirrors is asymmetric-Bethe-Ansatz-integrable, then any two of its $\delta$-function that cross at a dihedral angle $\pi/$(odd number) carry the same coupling constant, unless one or both mirrors are mirrors of an asymmetric reflection subgroup of the underlying reflection group. Such mirrors can be then assigned an infinite coupling constant.*

---

[1]Interestingly, for this particular choice of the coupling constants, the $\mathcal{G}$-generated generalized kaleidoscope is identical to the one generated by the $\tilde{A}_2$ reflection group ("red" mirrors alone). We would like to stress that the fact that $\mathcal{H} = \tilde{A}_1 \times \tilde{A}_1$ is not a reflection subgroup of $\tilde{A}_2$ has no significance. Here and in general, it is imperative that the group $\mathcal{H}$ is a *reflection subgroup* of the group $\mathcal{G}$ whose eigenstates are used to generate the eigenstates of the $\mathcal{G}$-$\mathcal{H}$-generated asymmetric generalized kaleidoscope.

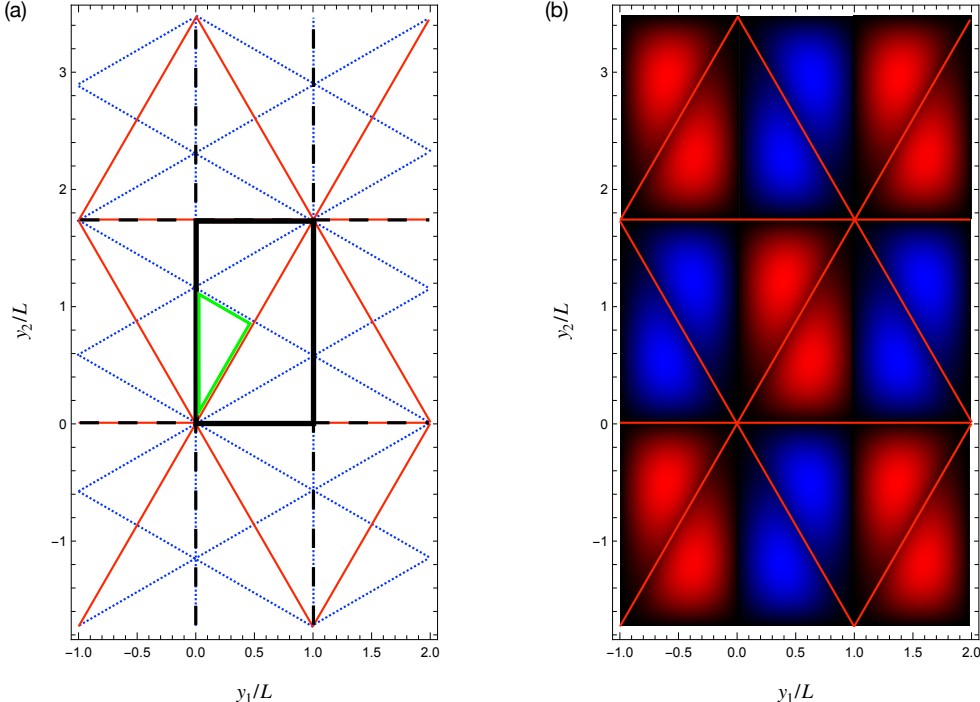

Figure 6: **The Liu-Qi-Zhang-Chen system (Fig. 5) as an alcove of a $\mathcal{G}$-$\mathcal{H}$-generated asymmetric generalized kaleidoscope.** Here, $\mathcal{G} = \tilde{G}_2$, $\mathcal{H} = \tilde{A}_1 \times \tilde{A}_1$. The boundaries of the alcoves of the $\mathcal{G}$ and $\mathcal{H}$ groups are represented by solid green and solid black lines respectively. (a) The $\delta$-function mirrors of the $\mathcal{G}$ group. For the Liu-Qi-Zhang-Chen system, the coupling constant on the "red" mirrors is the same as in the original problem (Fig. 5), and the "blue" coupling constant is set to zero. Black dashed lines are the mirrors of the $\mathcal{H}$ group; the solid-black-walled rectangle is its alcove. (b) Heat-map plot of the ground state wave function of the Liu-Qi-Zhang-Chen problem, computed numerically, positive (negative) values are shown in red (blue) color. According to the asymmetric Bethe Ansatz protocol, the set of the eigenstates of the problem can be found by identifying a subset of the eigenstates of the $\mathcal{G}$-kaleidoscope (red lines) that are odd with respect to the reflections about the mirrors of $\mathcal{H}$.

## 2.2 Physical manifestations of integrability of the Liu-Qi-Zhang-Chen system

In this subsection, we will present an example of an empirical manifestation of integrability of the Liu-Qi-Zhang-Chen system. It is common to use the level spacing statistics [10] as an integrability test. However, for the reasons outlined below, in this system, some traces of integrability survive even for large perturbations away from it, invalidating the level statistics method. Instead, we resort to a test for the primary features that deliver validity to the level spacing statistics analysis: the true and avoided crossing between the energy levels plotted as a function of an integrability-unrelated parameter [10].

We choose $\beta \equiv m_2/m_1$ as the integrability controlling parameter, and the coupling constant $g$ (the coefficient in front of the $\delta$-interaction $V(x_1 - x_2) = g\,\delta(x_1 - x_2)$) as the integrability-unrelated variable. Our primary object of interest is crossings between the levels of the same generic (i.e. present at all $\beta$) symmetry, common to both systems (and for that matter, for all $\beta$).

In Fig. 7(a), we show the energy spectrum of the asymmetric Bethe Ansatz integrable Liu-Qi-Zhang-Chen system, i.e. for mass ratio of tho particles equal to $\beta = 3$. It is instructive to

compare it to the spectrum of non-integrable counterpart shown in Fig. 7(b) where we use mass ratio equal to $\beta = \frac{5}{2} \times (1 + 1/(2 + 1/(3 + 1/(4 + \cdots))))) = 3.5828185668057793\ldots$. This specific choice of $\beta$ is being designed to represent a generic irrational number with manifestly unbounded continued fraction coefficients.

For any value of the mass ratio $\beta$, the system is invariant under a 180° rotation about the center of the corresponding rectangular billiard similar to the one represented in Fig. 5. In Fig. 7, red(black) energy lines correspond to the states that are even(odd) under this rotation.

In the absence of interactions ($g = 0$), the spectrum can be found explicitly for any mass ratio $\beta$ and is expressed in terms of two quantum numbers $n_1$ and $n_2$, according to

$$E_{n_1,n_2}^{g=0} = \frac{\beta n_1^2 + n_2^2}{\beta + 1}\mathcal{E}_0\,,$$
$$n_1 \geq 1\,, \quad n_2 \geq 1\,. \tag{1}$$

Here and below, $\mathcal{E}_0 = \frac{\pi^2 \hbar^2}{2\mu L^2}$ denotes the typical energy associated with zero-point fluctuations and $\mu \equiv \frac{m_1 m_2}{m_1 + m_2}$ the reduced mass. The noninteracting ground state energy is $E_{1,1}^{g=0} = \mathcal{E}_0$ independently of the specific value of $\beta$.

For impenetrable particles, $g = \infty$, the energy spectrum of the Liu-Qi-Zhang-Chen system is given by

$$E_{\tilde{n}_1,\tilde{n}_2}^{g=\infty} = \frac{\tilde{n}_1^2 + \tilde{n}_1 \tilde{n}_2 + \tilde{n}_2^2}{3}\mathcal{E}_0\,,$$
$$\tilde{n}_2 > \tilde{n}_1 \geq 1\,. \tag{2}$$

The ground state has an energy $E_{1,2}^{g=\infty} = \frac{7}{3}\mathcal{E}_0$. For the non-integrable counterpart, the $g = \infty$ spectrum is non-analytic, including the ground state. In both cases, the energy levels are doubly degenerate in $g \to \infty$ limit.

As we mentioned above, the Liu-Qi-Zhang-Chen system is asymmetric Bethe Ansatz integrable; it inherits additional integrals of motion of the underlying $\tilde{G}_2$ kaleidoscope (see Fig. 6). Thanks to the additional conserved quantities, levels of the same parity with respect to the 180° rotation (shown with the same colors) are allowed to cross. This is indeed what we observe in Fig. 7(a). The presence of such crossings is a manifestation of integrability.

In the non-integrable case, most of the crossings are lifted. The crossing $\mathbf{c}_5'$ (see Fig. 7(b)) constitutes a seeming exception. However, a much more accurate calculation reveals an avoided crossing. In the integrable case, this crossing remains a true one.

Explanation for the resilience of some of the crossings involves the interaction-insensitive eigenstates of the Liu-Qi-Zhang-Chen system, marked by $\mathbf{f}$ in Fig. 7(a). These states have a node along the interaction line. When the integrability is broken, the remnants of this node (marked $\mathbf{f'}$) survive, leading to a reduced coupling to other eigenstates.

In particular, the state $\mathbf{f'_{III}}$, non-integrable counterpart of the state $\mathbf{f_{III}}$ of the integrable system is responsible for the existence of the near-crossing $c_5'$ (see Fig. 7(b)). The Fig. 8 provides more details. This state is a linear combination of the three eigenstates of the noninteracting system (the black-walled rectangular $L \times (\sqrt{3}L)$ billiard (Fig. 8)), with the quantum numbers $(n_1, n_2) = \{(2, 8), (3, 7), (5, 1)\}$ (see (1)).

At the same time, the state $\mathbf{f_{III}}$ consists of two smoothly connected eigenstates of two similar right triangular $L \times (\sqrt{3}L)$ hard-wall billiards (the yellow-walled triangular billiard and its copy (Fig. 8)), with the quantum numbers $(\tilde{n}_1, \tilde{n}_2) = (1, 7)$. It is the even—with respect to a 180° rotation about the alcove center—eigenstate of the Liu-Qi-Zhang-Chen system at $g = \infty$; its energy spectrum is given by (2).

Finally, the state represented in this Figure is a tiling of six smoothly connected eigenstates of of six similar $L/\sqrt{3} \times L$ (the green-walled triangular billiard[2] and its five copies (Fig. 8)), with the quantum numbers $(\tilde{\tilde{n}}_1, \tilde{\tilde{n}}_2) = (2, 3)$. Any eigenstate of the green-walled billiard can be used to generate an interaction-insensitive eigenstate of the Liu-Qi-Zhang-Chen system. Hence, in general, the energies of the interaction-insensitive eigenstates will be given by

$$E^{g\text{-insensitive}}_{\tilde{\tilde{n}}_1, \tilde{\tilde{n}}_2} = (\tilde{\tilde{n}}_1^2 + \tilde{\tilde{n}}_1 \tilde{\tilde{n}}_2 + \tilde{\tilde{n}}_2^2) \mathcal{E}_0 \,,$$
$$\tilde{\tilde{n}}_2 > \tilde{\tilde{n}}_1 \geq 1 \,. \tag{3}$$

The node along the interaction line make the state energy be insensitive to the interactions. Deviations from the integrable mass ratio $\beta = 3$ will eventually populate the interaction line. However, the extremely weak splitting at the point $\mathbf{c}'_5$ in Fig. 6(b) indicates that even at $\beta = 3.58\ldots$ some traces of integrability at $\beta = 3$ remain. In particular, as we found numerically, the presence of near-crossings at $\beta \neq 3$ prevents formation of the zero-energy hole in the level spacing distribution making the deviations from integrability difficult to detect.

## 2.3 Asymmetric Bethe Ansatz: General considerations

The asymmetric BA method can be extended to any pair formed by a reflection group $\mathcal{G}$ and its reflection subgroup $\mathcal{H}$. Reflection subgroups of the reflection groups have been studied in detail in Ref [11]. Results of this article can be summarized as follows:

- Reflection subgroups of the finite reflection groups are listed in Chapter 3 of Ref. [11].

- For any undecomposable affine reflection group $\mathcal{G}$ and for any integer scaling factor $d$, a reflection subgroup $\mathcal{H}$ can be found whose alcove is a homothetic copy of the alcove of $\mathcal{G}$, with a dilation factor $d$ (Lemma 9 of [11], see Fig. 9(a) as an example).

- If a group $\mathcal{G}$ is decomposable onto a product of smaller reflection groups, the corresponding reflection subgroup $\mathcal{H}$ can be a product of dilation subgroups of the components of $\mathcal{G}$, with, generally, different dilation factors (Fig. 9(b)).

- Affine reflection groups $\mathcal{G} = \tilde{C}_2, \tilde{G}_2, \tilde{F}_4$ have reflection subgroups $\mathcal{H}$ with alcoves that are similar to the ones for $\mathcal{G}$ but not homothetic to them (Table 2 of Ref. [11], see Fig. 9(c) as an example).

- There are several potentially important cases where the alcove of $\mathcal{H}$ is not similar to the one of $\mathcal{G}$. $\mathcal{H}$ can be either an indecomposable (Table 3 of Ref. [11]) or a decomposable reflection subgroup (Table 5 of Ref. [11], see Fig. 9(d) as an example; also this is the example that solves the Liu-Qi-Zhang-Chen problem [9][3]). Here $\mathcal{G}$ is an indecomposable affine reflection group in this context.

- An alcove of $\mathcal{H}$ may have an infinite volume (Theorem 3 of Ref. [11], see Fig. 9(e) as an example).

- All the techniques described in Ref. [11] may be combined with one another. Figure 9(f) shows an example of a combination of a non-similar subgroup with a decomposable homothety.

---

[2]This billiard is also an alcove for the reflection group $\tilde{G}_2$ that is used to generate solutions of the Liu-Qi-Zhang-Chen problem.

[3]The kaleidoscopes of Fig. 9(d) and Fig. 6(a) are mirror images of each other, with respect to a mirror at 45° to the horizon.

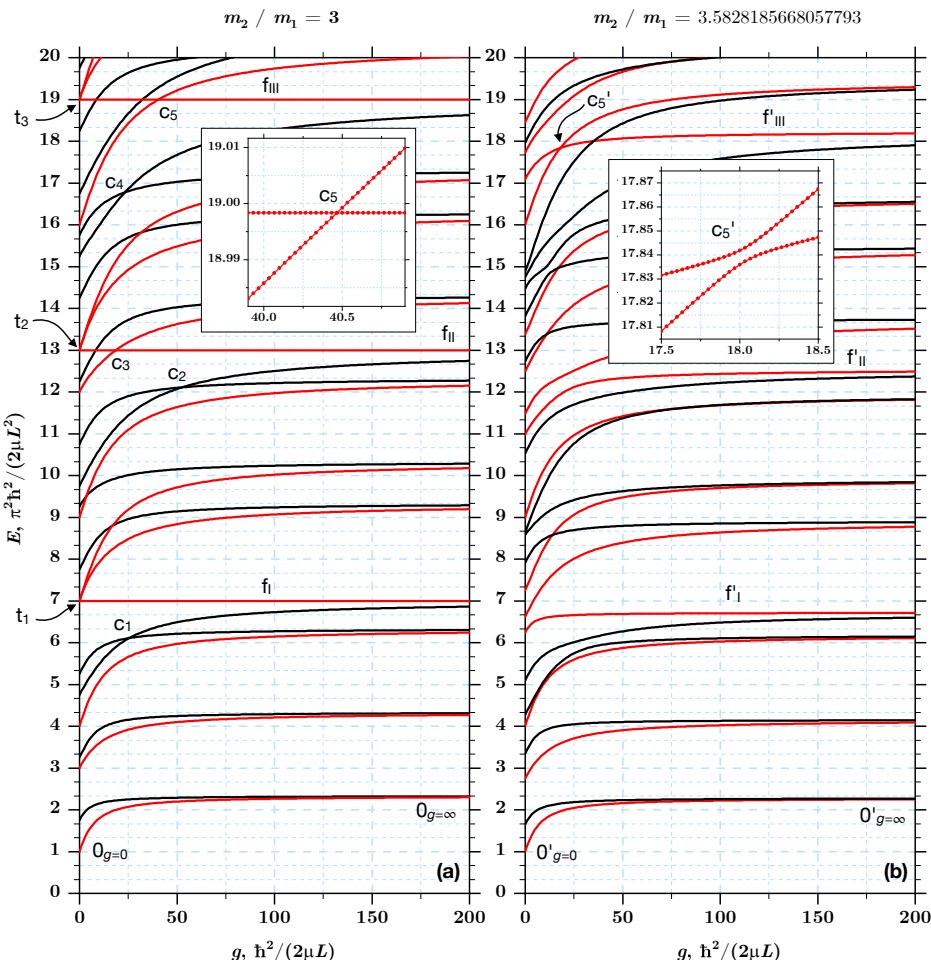

Figure 7: **Energy levels as a function of the coupling constant, for two $\delta$-interacting particles in a hard-wall box of length $L$.** (a) Mass ratio $m_1$:$m_2$ is 1:3 for Subfigure (a) and 1:3.5828185668057793...for Subfigure (b). The former corresponds to the Liu-Qi-Zhang-Chen system represented in Fig. 5. Red(black) lines correspond to the states that are even(odd) under the 180° rotation about the center of a rectangular billiard similar to the one presented in Fig. 5. Labels $\mathbf{0_{g=0}}$ and $\mathbf{0'_{g=0}}$ mark the non-interacting integrable and non-integrable ground state energies respectively. Labels $\mathbf{0_{g=\infty}}$ and $\mathbf{0'_{g=\infty}}$ denote the integrable and non-integrable ground state energies in the hard-core regime respectively. The crossings between the same-parity energy levels are marked with $\mathbf{t}$ and $\mathbf{c}$ labels. The $\mathbf{t}$ crossings are triply-degenerate levels of the noninteracting system each of which contain an interaction-insensitive eigenstate of an energy (3). This state is a linear combination of the three eigenstates of the noninteracting system (1) with $(n_1, n_2) = \left\{ (\tilde{\tilde{n}}_1, \tilde{\tilde{n}}_1 + 2\tilde{\tilde{n}}_2), (\tilde{\tilde{n}}_2, 2\tilde{\tilde{n}}_1 + \tilde{\tilde{n}}_2), (\tilde{\tilde{n}}_2 + \tilde{\tilde{n}}_1, \tilde{\tilde{n}}_2 - \tilde{\tilde{n}}_1) \right\}$. The $\mathbf{c}$ crossings appear at finite values of coupling. Insets in (a) and (b) magnify the regions in the vicinity of $\mathbf{c_5}$ and $\mathbf{c'_5}$ points respectively. In spite of its appearance, the crossing $\mathbf{c'_5}$ is indeed an avoided crossing, while the $\mathbf{c_5}$ is a true one, as expected from the integrability considerations. The energy levels $\mathbf{f}$ and $\mathbf{f'}$ are the interaction-insensitive and near-interaction-insensitive eigenstates of the integrable and non-integrable systems respectively.
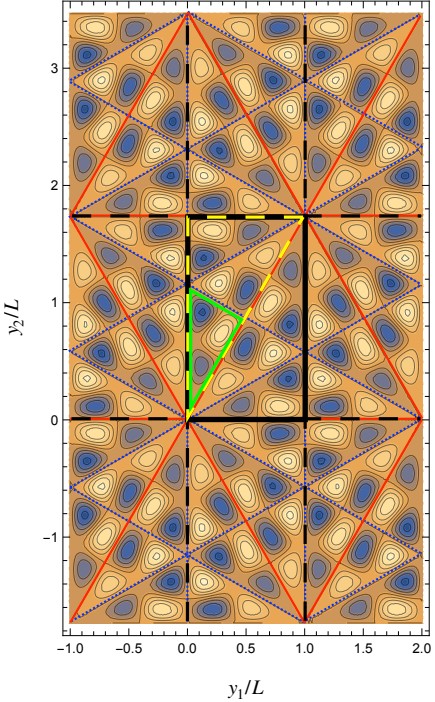

Figure 8: **An example of an interaction-insensitive eigenstate of the Liu-Qi-Zhang-Chen system.** In Fig. 7(a), the energy of this state is marked by $\mathbf{f}'_{\mathrm{III}}$. This state explains the resilience of the level crossing $\mathbf{c}'_5$ in Fig. 7(b). Any interaction-insensitive state is a smooth sign-alternating tiling of the six identical eigenstates of the green-walled billiard. Any eigenstate of the green-walled billiard can be used to generate this tiling. (Curiously, these eigenstates are also the eigenstates of the hard-core version of the conventional generalized kaleidoscope $\tilde{G}_2$ that was used to generate solutions of the Liu-Qi-Zhang-Chen problem (see Fig. 6(a)).) Because of the absence of cusp, this tiling is naturally an eigenstate of the free system. Because of the node along the interaction line (red), adding interaction does not affect the eigenstate. As expected, a tiling of three out of six green-walled billiard eigenstates is also an eigenstates of the yellow-walled billiard relevant to the hard-core interaction case.

## 2.4 A worked example: A spatially odd excited state of a bosonic dimer in a $\delta$-well

As was mentioned before, this article had been inspired by the failure of the conventional Bethe Ansatz to explain integrability of the Liu-Qi-Zhang-Chen system [9]. There, obtaining its eigenenergies and eigenfunctions would still require solving the so-called Bethe Ansatz equations [1] numerically. This is because in that problem, the particles' motion is confined to a finite region of space. In order to better illustrate the strength of the asymmetric BA method, we chose the system of two $\delta$-interacting bosons in a $\delta$ potential in the spatially odd subspace of the Hilbert space [12]. Here, the energy spectrum and the eigenstates can be obtained analytically, with the use of the asymmetric BA. The scattering eigenstates of this problem were analyzed in [12]. However, its bound states have never obtained in literature. Below, we accomplish this task.

Consider a reflection group $C_2$ [4] that constitutes the full symmetry group of a square. In a 2D plane $x_1 x_2$, it can be generated by two reflections, e.g. by $(x_1, x_2) \rightarrow (-x_1, x_2)$

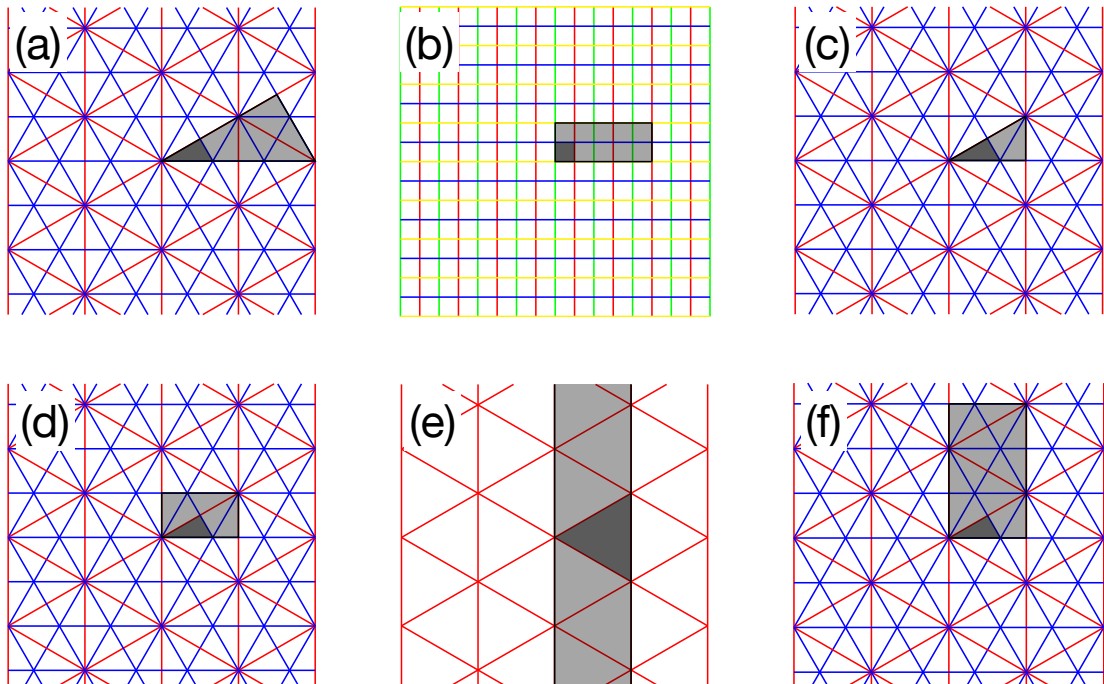

Figure 9: **Several significant features of the complete list of reflection subgroups $\mathcal{H}$ of indecomposable reflection groups $\mathcal{G}$ presented in [11].** Here, a deep-grey polygon represents a sample alcove of the corresponding indecomposable reflection groups $\mathcal{G}$; a light-grey polygon is an alcove of the corresponding reflection subgroup $\mathcal{H}$ of $\mathcal{G}$. (a) All indecomposable affine reflection groups have reflection subgroups whose alcoves are homothetic copies of the original group. (b) Decomposable affine reflection groups may several distinct dilution factors. (c) There are several cases when the $\mathcal{H}$ alcove is similar to the $\mathcal{G}$ alcove but not homothetic to it. (d) In several cases, the $\mathcal{H}$ alcove has a shape that is completely different from the shape of the $\mathcal{G}$ alcove. (e) $\mathcal{H}$ alcove may have an infinite volume. (f) Methods for generating the reflection subgroups $\mathcal{H}$ described in [11] can be combined.

and $(x_1, x_2) \rightarrow (x_2, x_1)$. The whole group consists of eight elements: this list is comprised of an identity, three rotations: $(x_1, x_2) \rightarrow (-x_2, x_1)$, $(x_1, x_2) \rightarrow (-x_1, -x_2)$, and $(x_1, x_2) \rightarrow (x_2, -x_1)$, and four reflections: $(x_1, x_2) \rightarrow (-x_1, x_2)$, $(x_1, x_2) \rightarrow (x_1, -x_2)$, $(x_1, x_2) \rightarrow (x_2, x_1)$, and $(x_1, x_2) \rightarrow (-x_2, -x_1)$.

Consider a two-particle Hamiltonian that is invariant under any of the elements of the $C_2$ group.

$$\hat{H}_{C_2} = -\frac{\hbar^2}{2m}\left(\frac{\partial^2}{\partial x_1^2} + \frac{\partial^2}{\partial x_2^2}\right) + g_B\delta(x_1) + g_B\delta(x_2) + g\delta(x_1 - x_2) + g\delta(x_1 + x_2), \qquad (4)$$

where $m$ is the particle mass, and $g_B$ and $g$ are the particle-barrier and particle-particle coupling constants respectively. In addition to the three empirically relevant interactions — two particle-barrier and one particle-particle — the Hamiltonian (4) contains a nonlocal, unphysical $\delta(x_1 + x_2)$ term. A priori, such observation indicates a complete physical irrelevance of the Hamiltonian in question. However, below we will apply the asymmetric Bethe Ansatz method to identify a sector of the Hilbert space that shares its eigenstates with eigenstates of a system of two $\delta$-interacting bosons in a field of a $\delta$-barrier (or well). We will focus on the bound states. Note that the scattering states for the same problem have been considered in [12].

According to the asymmetric Bethe Ansatz recipe, we need to first construct a Bethe Ansatz state that is not even with respect to the future perfectly reflective mirror. In our case, the mirror in question is the one that generates the $(x_1, x_2) \rightarrow (-x_2, -x_1)$ reflection.

Following the standard Bethe Ansatz prescription [1, Sec. 5.2], we will assume that in each of the eight domains separated by the $\delta$-barriers,

$$x_1 \geq 0 \wedge x_1 - x_2 \leq 0,$$
$$x_1 \geq 0 \wedge x_1 + x_2 \leq 0,$$
$$x_1 \leq 0 \wedge x_1 + x_2 \geq 0,$$
$$x_1 \leq 0 \wedge x_1 - x_2 \geq 0,$$
$$x_2 \geq 0 \wedge x_2 - x_1 \leq 0,$$
$$x_2 \geq 0 \wedge x_1 + x_2 \leq 0,$$
$$x_2 \leq 0 \wedge x_1 + x_2 \geq 0,$$
$$x_2 \leq 0 \wedge x_2 - x_1 \geq 0,$$

the eigenstate wavefunction, $\psi(x_1, x_2)$, is a superposition of plane waves

$$e^{i\vec{k}\cdot\vec{x}},$$

(with a potentially imaginary wavevector) with wavevectors that are group images of a "seed" wavevector. For a bound state, we will expect the wavevectors to be purely imaginary, i.e.

$$\vec{k} = -i\vec{\kappa},$$

with

$$\vec{\kappa} = \text{real}.$$

The coefficients of the superposition are allowed to undergo a discontinuity at the domain boundaries. The resulting boundary conditions read

$$\psi(x_1, x_2) \stackrel{x_1 \rightarrow 0}{\approx} A_{\mathrm{B},1}(x_2)\left(1 - |x_1|/a_{\mathrm{B}} + \mathcal{O}(x_1^2)\right),$$
$$\psi(x_1, x_2) \stackrel{x_2 \rightarrow 0}{\approx} A_{\mathrm{B},2}(x_1)\left(1 - |x_2|/a_{\mathrm{B}} + \mathcal{O}(x_2^2)\right),$$
$$\psi(x_1, x_2) \stackrel{x_1 \rightarrow x_2}{\approx} A_-((x_1+x_2)/2)\left(1 - |x_1 - x_2|/a + \mathcal{O}((x_1 - x_2)^2)\right),$$
$$\psi(x_1, x_2) \stackrel{x_1 \rightarrow -x_2}{\approx} A_+((x_1-x_2)/2)\left(1 - |x_1 + x_2|/a + \mathcal{O}((x_1 + x_2)^2)\right),$$

$$(5)$$

where the particle-barrier and particle-particle scattering lengths $a_{\mathrm{B}}$ and $a$ are given by

$$a_{\mathrm{B}} \equiv -\frac{\hbar}{mg_{\mathrm{B}}},$$
$$a \equiv -\frac{\hbar}{\mu g_{\mathrm{B}}},$$

with $\mu = m/2$ being a reduced mass. For certainty, we will assume that the eigenstate that we are building is of a "scattering type," i.e. that in one of the domains (the "outgoing wave" domain), only one plane wave is present. In addition, we will require that the state $\psi(x_1, x_2)$ is even with respect to the $(x_1, x_2) \rightarrow (x_2, x_1)$ reflection. We will choose the $x_1 \geq 0 \wedge x_1 - x_2 \leq 0$ as the "outgoing wave" domain.

In the parameter range

$$(a > 0 \wedge a_{\mathrm{B}} > 0) \vee (a > 0 \wedge a_{\mathrm{B}} < -a) \vee (a_{\mathrm{B}} > 0 \wedge a < -2a_{\mathrm{B}}),$$

one finds the ground bound state of the system, with an energy

$$E_0 = -\frac{\hbar^2}{m}\frac{a^2 + 2aa_B + 2a_B^2}{a^2 a_B^2}\,,$$

and wavevectors

$$(\vec{\kappa}_0)_{\sigma_1,\sigma_2}^\sigma = \begin{cases} \left(\sigma_1(-\frac{2}{a}-\frac{1}{a_B}),\, \sigma_2(-\frac{1}{a_B})\right), & \text{for}\quad \sigma = +\,, \\ \left(\sigma_2(-\frac{1}{a_B}),\, \sigma_1(-\frac{2}{a}-\frac{1}{a_B}))\right), & \text{for}\quad \sigma = -\,, \end{cases}$$

where $\sigma = +/-$, $\sigma_1 = +/-$, $\sigma_2 = +/-$. This state can be shown to be even with respect to all the reflections of the group and as such, of no interest.

The corresponding (unnormalized) wavefunction is

$$\psi_0^{(\text{BA})}(x_1, x_2) = \frac{1}{a_B}\frac{\sqrt{(\chi+2)(\chi+1)}}{|\chi|} \times$$

$$\begin{cases} e^{(\vec{\kappa}_0)_{(++)}^{(-)}\cdot\vec{x}}, & x_1 \geq 0 \wedge x_1 - x_2 \leq 0\,, \\ e^{(\vec{\kappa}_0)_{(-+)}^{(-)}\cdot\vec{x}}, & x_1 \geq 0 \wedge x_1 + x_2 \leq 0\,, \\ e^{(\vec{\kappa}_0)_{(+-)}^{(-)}\cdot\vec{x}}, & x_1 \leq 0 \wedge x_1 + x_2 \geq 0\,, \\ e^{(\vec{\kappa}_0)_{(--)}^{(-)}\cdot\vec{x}}, & x_1 \leq 0 \wedge x_1 - x_2 \geq 0\,, \\ e^{(\vec{\kappa}_0)_{(++)}^{(+)}\cdot\vec{x}}, & x_2 \geq 0 \wedge x_2 - x_1 \leq 0\,, \\ e^{(\vec{\kappa}_0)_{(-+)}^{(+)}\cdot\vec{x}}, & x_2 \geq 0 \wedge x_1 + x_2 \leq 0\,, \\ e^{(\vec{\kappa}_0)_{(+-)}^{(+)}\cdot\vec{x}}, & x_2 \leq 0 \wedge x_1 + x_2 \geq 0\,, \\ e^{(\vec{\kappa}_0)_{(--)}^{(+)}\cdot\vec{x}}, & x_2 \leq 0 \wedge x_2 - x_1 \geq 0\,, \end{cases} \tag{6}$$

where

$$\chi \equiv \frac{a}{a_B}\,,$$

In a narrower parameter range,

$$0 < \frac{a}{2} < a_B < a\,,$$

the first (and the only) excited bound state (three-fold degenerate) manifold can be found, with an energy

$$E_1 = -\frac{\hbar^2}{m}\frac{a^2 - 2aa_B + 2a_B^2}{a^2 a_B^2}\,,$$

and the wavevectors

$$(\vec{\kappa}_1)_{\sigma_1,\sigma_2}^\sigma = \begin{cases} \left(\sigma_1(\frac{2}{a}-\frac{1}{a_B}),\, \sigma_2(-\frac{1}{a_B})\right), & \text{for}\quad \sigma = +\,, \\ \left(\sigma_2(-\frac{1}{a_B}),\, \sigma_1(\frac{2}{a}-\frac{1}{a_B}))\right), & \text{for}\quad \sigma = -\,. \end{cases}$$

Its (normalized) wavefunction has a form (Fig.10(a))

$$\psi_1^{(BA)}(x_1, x_2) = \frac{1}{a_B} \frac{1}{\sqrt{(7 - (\chi - 1)^2)(\chi - 1)\chi}}$$

$$\times \begin{cases} (\chi - 1)(\chi - 2)e^{(\vec{\kappa}_1)_{(++)}^{(+)} \cdot \vec{x}}, & x_1 \geq 0 \wedge x_1 - x_2 \leq 0, \\ (\chi - 1)\chi e^{(\vec{\kappa}_1)_{(+-)}^{(+)} \cdot \vec{x}} - 2(\chi - 2)e^{(\vec{\kappa}_1)_{(++)}^{(-)} \cdot \vec{x}} - 2e^{(\vec{\kappa}_1)_{(--)}^{(+)} \cdot \vec{x}}, & x_1 \geq 0 \wedge x_1 + x_2 \leq 0, \\ (\chi - 1)\left( \chi e^{(\vec{\kappa}_1)_{(-+)}^{(+)} \cdot \vec{x}} - 2e^{(\vec{\kappa}_1)_{(++)}^{(+)} \cdot \vec{x}} \right), & x_1 \leq 0 \wedge x_1 + x_2 \geq 0, \\ (\chi + 2)(\chi - 1)e^{(\vec{\kappa}_1)_{(--)}^{(+)} \cdot \vec{x}} - 2(\chi - 2)e^{(\vec{\kappa}_1)_{(+-)}^{(-)} \cdot \vec{x}} - 2\chi e^{(\vec{\kappa}_1)_{(+-)}^{(+)} \cdot \vec{x}}, & x_1 \leq 0 \wedge x_1 - x_2 \geq 0, \\ (\chi - 1)(\chi - 2)e^{(\vec{\kappa}_1)_{(++)}^{(-)} \cdot \vec{x}}, & x_2 \geq 0 \wedge x_2 - x_1 \leq 0, \\ (\chi - 1)\chi e^{(\vec{\kappa}_1)_{(+-)}^{(-)} \cdot \vec{x}} - 2(\chi - 2)e^{(\vec{\kappa}_1)_{(++)}^{(+)} \cdot \vec{x}} - 2e^{(\vec{\kappa}_1)_{(--)}^{(-)} \cdot \vec{x}}, & x_2 \geq 0 \wedge x_1 + x_2 \leq 0, \\ (\chi - 1)\left( \chi e^{(\vec{\kappa}_1)_{(-+)}^{(-)} \cdot \vec{x}} - 2e^{(\vec{\kappa}_1)_{(++)}^{(-)} \cdot \vec{x}} \right), & x_2 \leq 0 \wedge x_1 + x_2 \geq 0, \\ (\chi + 2)(\chi - 1)e^{(\vec{\kappa}_1)_{(--)}^{(-)} \cdot \vec{x}} - 2(\chi - 2)e^{(\vec{\kappa}_1)_{(+-)}^{(+)} \cdot \vec{x}} - 2\chi e^{(\vec{\kappa}_1)_{(+-)}^{(-)} \cdot \vec{x}}, & x_2 \leq 0 \wedge x_2 - x_1 \geq 0. \end{cases} \quad (7)$$

It can be shown that $\psi_1^{(BA)}(x_1, x_2)$, $\psi_1^{(BA)}(-x_2, -x_1)$, and $\psi_1^{(BA)}(-x_2, x_1)$ are linearly independent, but action of any other element of $C_2$ on $\psi_1^{(BA)}(x_1, x_2)$ will lead to a linear combination of these three states. Hence, $\psi_1^{(BA)}(x_1, x_2)$, $\psi_1^{(BA)}(-x_2, -x_1)$, and $\psi_1^{(BA)}(-x_2, x_1)$ form a three-dimensional representation of $C_2$. We will see below that this representation can be further reduced to a product of a one-dimensional and a two-dimensional irreducible representations.

As a preparatory step, let us compute the overlap $\Upsilon$ integral between $\psi_1^{(BA)}(x_1, x_2)$ and its reflection about the $x_1 = -x_2$ mirror:

$$\Upsilon \equiv \int dx_1 dx_2 \psi_1^{(\text{asymmetric BA})}(x_1, x_2) \psi_1^{(\text{asymmetric BA})}(-x_2, -x_1) = -\frac{(\chi - 2)^2}{7 - (\chi - 1)^2}. \quad (8)$$

Now, an even (normalized) linear combination, $\frac{1}{\sqrt{2}\sqrt{1+\Upsilon}}(\psi_1^{(BA)}(x_1, x_2) + \psi_1^{(BA)}(-x_2, -x_1))$, can be shown to form an identity representation of $C_2$.

The odd combination (Fig.10(b))

$$\psi_1^{(\text{Asymmetric BA})}(x_1, x_2) = \frac{1}{\sqrt{2}\sqrt{1-\Upsilon}}(\psi_1^{(BA)}(x_1, x_2) - \psi_1^{(BA)}(-x_2, -x_1)), \quad (9)$$

i.e. the second step of the asymmetric Bethe Ansatz procedure, produces the state we are looking for. It has a node along the $x_1 = -x_2$ line and as such, it is an eigenstate of a problem where the $g_B\delta(x_1 + x_2)$ interaction in the Hamiltonian (4) is replaced by a perfectly reflective mirror. It may be conjectured that all the eigenstates of the latter problem can be obtained using the asymmetric Bethe Ansatz.

It is also the case that (9) is an $((x_1, x_2) \to (-x_2, -x_1))$-odd eigenstate of a problem with *any* value of coupling in front of $\delta(x_1 + x_2)$, including no coupling at all. As such, is spatially odd excited bound state of a bosonic dimer trapped in the field of a $\delta$-well. Again, the scattering version of this problem has been considered in [12].

Finally, one can show that $\psi_1^{(\text{asymmetric BA})}(x_1, x_2)$ and $\psi_1^{(\text{asymmetric BA})}(-x_2, x_1)$ form a two-dimensional representation of the group $C_2$. These two states are orthogonal to each other.

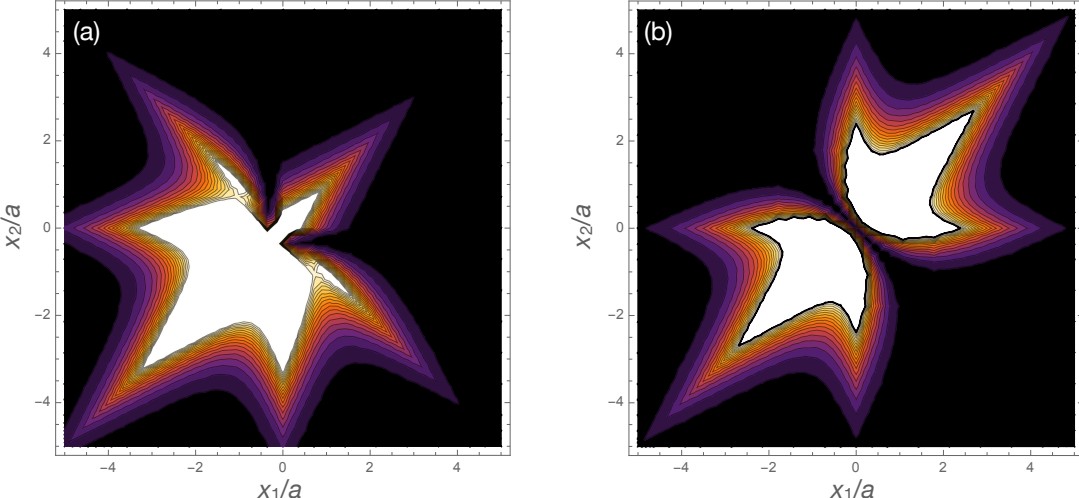

Figure 10: **A standard Bethe Ansatz eigenstate and a generated thereby asymmetric Bethe Ansatz eigenstate within the the first excited bound state of the $C_2$ kaleidoscope (see text): contour plot of the density.** Scattering length of the $\delta$-barrier is $\frac{3}{4}a$. The energy of the (triply-degenerate) excited bound manifold is $E_1 = -\frac{10}{9}\hbar^2/ma^2$. For comparison, for the same set of parameters, the ground state energy is $E_0 = -\frac{58}{9}\hbar^2/ma^2$. (a) A "natural" Bethe Ansatz eigenstate: a bound state whose structure is inspired by a scattering state with the outgoing "wave" occupying the top right quadrant. (b) An asymmetric Bethe Ansatz eigenstate obtained using antisymmetrization of the subfigure (a) state with respect to the $x_1 = -x_2$ diagonal. This state can serve as an eigenstate of any modification of the $C_2$ Hamiltonian where the strength of the $x_1 = -x_2$ $\delta$-barrier or well can be chosen at will. In particular, no barrier at all corresponds to two bosons in a field of a stationary $\delta$-well [12].

For future reference, let us list the asymmetric-Bethe-Ansatz-inspired eigenstates of the first excited manifold:

1-dimensional irreducible representation of $C_2$:

$$\frac{1}{\sqrt{2}\sqrt{1+\Upsilon}}(\psi_1^{(BA)}(x_1, x_2) + \psi_1^{(BA)}(-x_2, -x_1)),$$

even under all eight members of $C_2$.

2-dimensional irreducible representation of $C_2$:

$$\psi_1^{(\text{asymmetric BA})}(x_1, x_2) \equiv \frac{1}{\sqrt{2}\sqrt{1-\Upsilon}}(\psi_1^{(BA)}(x_1, x_2) - \psi_1^{(BA)}(-x_2, -x_1)),$$

odd under $(x_1, x_2) \rightarrow (-x_2, -x_1)$; even under $(x_1, x_2) \rightarrow (x_2, x_1)$,

$$\psi_1^{(\text{asymmetric BA})}(-x_2, x_1),$$

even under $(x_1, x_2) \rightarrow (-x_2, -x_1)$; odd under $(x_1, x_2) \rightarrow (x_2, x_1)$.

## 3 Summary and outlook

In this article, we generalize the conventional Bethe Ansatz, by breaking the symmetry $\mathcal{G}$ of the generalized kaleidoscope the Ansatz is based on. The new method allows to replace a set of semitransparent mirrors of the original generalized kaleidoscope by reflecting mirrors. The set

of such modified mirrors must coincide with mirrors of a reflection subgroup $\mathcal{H}$ of the original reflection group $\mathcal{G}$. One of the consequences of this generalization is that the mirrors crossing at an angle $\pi/(\text{odd number})$ are no longer required to have the same coupling constant. We decided to call the new exact solution method the asymmetric Bethe Ansatz (asymmetric BA).

As a worked example, we construct the specially odd bound state of a bosonic dimer trapped in a $\delta$-well, using an asymmetric Bethe Ansatz extension of the Bethe Ansatz solution for the $C_2$ generalized kaleidoscope.

The next step will be to identify the *previously unknown, empirically relevant instances of the asymmetric BA*, besides the Liu-Qi-Zhang-Chen that our paper studied in detail. One already known instance is the problem of scattering of two one-dimensional $\delta$-interacting bosons on a $\delta$-barrier: it was found to be solvable in the spatially odd sector of the Hilbert space [12] (see [13] for a lattice analogue of this problem). There, $\mathcal{G} = C_2$ and $\mathcal{H} = A_1$ represented by a mirror along the second diagonal (see Ref. [11], Lemma 2, Corollary 1). Another known example is $\mathcal{G} = \tilde{A}_{N-1}$ and $\mathcal{H} = \tilde{A}_{N_1-1} \times \tilde{A}_{N_2-1}$, with $N_1 + N_2 = N$ (where $\tilde{A}_1$ is associated with $\tilde{I}_1$, and $\tilde{A}_0$ is associated with the trivial group). It can be regarded as a $N_1$ spin-up/$N_2$ spin-down sector of the Yang-Gaudin model of spin-1/2 one-dimensional $\delta$-interacting fermions [7,8], on a ring; it's bosonized version has been studied in Ref. [14]. This case is covered by Theorem 3 of Ref. [11]. Figure 9(e) provides an example for $N = 3 = 1 + 2$. Additionally, the ($\mathcal{G} = \tilde{C}_N$, $\mathcal{H} = \tilde{C}_{N_1} \times \tilde{C}_{N_2}$ assignment leads to the Yang-Gaudin model in a hard-wall box (Table 5 of Ref. [11]).

# Acknowledgments

We are immeasurably grateful to Vanja Dunjko, Anna Minguzzi, Patrizia Vignolo, and Yunbo Zhang for numerous discussions.

**Funding information** M.O. was supported by the NSF Grant No. PHY-1912542. G.E.A. acknowledges support by the Spanish Ministerio de Ciencia e Innovación (MCIN/AEI/ 10.13039/501100011033, grant PID2020-113565GB-C21), and by the Generalitat de Catalunya (grant 2021 SGR 01411).

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
