# Peer review of "Asymmetric Bethe Ansatz"

_SciPost Physics Core, doi:SciPost Phys. Core 7, 062 (2024)_

## Round 2 · Referee Report · Anonymous (Referee 1) · 2024-8-14

Report

The authors made major revisions to the paper. The paper has improved considerably. Now it is much more readable, more clear, the introduction is much better and the worked out example helps a lot.

I have only minor questions remaining.

And despite the significant improvements, my recommendation is to publish in SciPost Physics Core. Reasons for this being that I see fewer researchers interested in this line of developments, and that the progress given in the paper is not significant enough to get into SciPost Physics. According to my judgement, if there is some progress in a topic which is very significant for current research, then it should be easier to get into SciPost Physics or other high journal paper. The scientific progress in this paper is fairly good (not groundbreaking, but also, not minimal), but the topic seems to be of less interest for the community today.

So the final recommendation is:

This submission does not meet the criteria of SciPost Physics, but does meet those of SciPost Physics Core, where it could be published. (even without further review, trusting that the authors consider the minor questions below)

Requested changes

1- There are still one or two typos, for example kaleidoscope was misspelled once. 2- Please explain in the text, what is the reason for choosing the "worked example". Do I understand it correctly, that this is NOT the the Liu-Qi-Zhang-Chen problem?

Recommendation

Accept in alternative Journal (see Report)

  • validity: top
  • significance: good
  • originality: high
  • clarity: top
  • formatting: perfect
  • grammar: excellent

Author:  Maxim Olshanii  on 2024-08-19  [id 4699]

(in reply to Report 1 on 2024-08-14)

We indeed grateful to the referee for forcing our manuscript to become more readable.

We agree that SciPost Core is a better venue.

The misprints have been corrected.

We added a note explaining the choice for the worked example.

---

## Round 2 · Referee Report · Anonymous (Referee 2) · 2024-8-22

Report

I went through the revised manuscript submitted by the authors . The revised introduction and the example of the bosonic dimer in Sec. 2,4 help the reader understand the context and the applicability of Asymmetric Bethe Ansatz. However, even after revision by the Authors, I believe that the manuscript is more suited for a specialized audience since it has a high level of technicalities.
For this reason, I would recommend the publication of this manuscript in SciPost Physics Core.

Recommendation

Accept in alternative Journal (see Report)

  • validity: good
  • significance: good
  • originality: top
  • clarity: high
  • formatting: -
  • grammar: -

Author:  Maxim Olshanii  on 2024-08-23  [id 4713]

(in reply to Report 2 on 2024-08-22)

We do agree, SciPost Physics Core is a more suitable venue. And thank you for your valuable time and energy.

---

## Round 2 · Author Response

We are grateful to the referees: both of them were true experts on the subject . Remarks from both referees led to a whole new study: the one that describes bound states of the C_{2} model in all quantitative details. It is an ideal setting to illustrate the Asymmetric BA idea, as suggested by the referees. New version of our manuscript is 30% longer. We agreed with all referees' objections save one (on the best empirical manifestation of integrability to be presented).

---

## Round 2 · List of Changes

• A whole new section (Sec. 2.4) (5 pages + 1 figure) is added.

  • We also completely reworked the abstract

  • We added a "pedestrian" section (Sec. 1.1) to Introduction.

---

## Round 4 · Author Response

We are truly grateful to both the Editors and the referees for investing their valuable time and energy into making our paper better. We do agree that SciPost Core is the most suitable venue for our manuscript.

---

## Round 4 · List of Changes

• New section on an explicit derivation for spatially odd bound states for two $\delta$ interacting bosons in a $\delta$ potential.
  • A pedestrian explanation for how the conventional Bethe Ansatz works.

---

## Editorial Decision

published